# In vivo simultaneous nonlinear absorption Raman and fluorescence (SNARF) imaging of mouse brain cortical structures

Andrew T. Francis[1], Bryce Manifold [1], Elena C. Carlson[1], Ruoqian Hu[1], Andrew H. Hill[1], Shuaiqian Men[1] & Dan Fu [1✉]

Label-free multiphoton microscopy is a powerful platform for biomedical imaging. Recent advancements have demonstrated the capabilities of transient absorption microscopy (TAM) for label-free quantification of hemoglobin and stimulated Raman scattering (SRS) microscopy for pathological assessment of label-free virtual histochemical staining. We propose the combination of TAM and SRS with two-photon excited fluorescence (TPEF) to characterize, quantify, and compare hemodynamics, vessel structure, cell density, and cell identity in vivo between age groups. In this study, we construct a simultaneous nonlinear absorption, Raman, and fluorescence (SNARF) microscope with the highest reported in vivo imaging depth for SRS and TAM at 250–280 µm to enable these multimodal measurements. Using machine learning, we predict capillary-lining cell identities with 90% accuracy based on nuclear morphology and capillary relationship. The microscope and methodology outlined herein provides an exciting route to study several research topics, including neurovascular coupling, blood-brain barrier, and neurodegenerative diseases.

[1] Department of Chemistry, University of Washington, Seattle, WA 98195, USA. ✉email: danfu@uw.edu

The mouse brain is a complex organ that contains over 100 million cells[1] that collectively consume ~20% of bodily energy at rest[2]. Understanding the underlying cell-to-cell and cell-to-vasculature interactions remains paramount to studying the brain under physiological and pathological states. Multiphoton microscopy affords an attractive route to probe these interactions through subcellular spatial resolution and intrinsic optical sectioning. Two-photon excited fluorescence (TPEF) has emerged as a leading multiphoton imaging technique due to these benefits as well as advancements in fluorescent transgenic animals and highly specific fluorescent dyes. However, the requirement of labeling brings limitations: (1) TPEF emission spectra are typically broad (~50 nm), which limits the number of separable fluorescent contrasts, and thus, information that can be collected from one animal, (2) generating animals with multiple transgenes can be challenging, time-consuming, and expensive[3], and (3) application of fluorescent dyes— topically or intravenously—is invasive which can alter the animal's physiology[4].

To circumvent these limitations, orthogonal label-free multiphoton imaging techniques can generate additional image contrasts. Third harmonic generation (THG) is one such method that provides characterization of myelination and cell density in the brain[5,6]. While robust, THG does not provide molecular information. Instead, researchers have explored the use of coherent Raman scattering (CRS) microscopy for label-free chemical imaging. Coherent anti-Stokes Raman scattering (CARS) and stimulated Raman scattering (SRS)—two CRS techniques—have shown tremendous capability in the chemical mapping of endogenous lipids, proteins, and other biomolecules in animal and human tissues[7–12]. SRS has several advantages over CARS, including a linear dependence between signal size and chemical concentration and the absence of a non-resonant background, making SRS the more widely employed of the two CRS techniques. One emerging application of SRS microscopy is the development of label-free histopathology, which achieves comparable diagnostic insight as traditional staining methods (e.g., H&E)[13,14]. Despite its capabilities, reports of in vivo brain imaging with SRS remains scarce largely due to poor penetration depth. Ji et al. reported an SRS imaging depth of ~100 μm when demonstrating in vivo pathological differentiation between tumor and non-neoplastic tissue in a xenografted mouse brain model[9]. To address this issue, our group and others have worked to increase signal sizes and thus penetration depth, via pulse optimization[15], wavelength optimization[16], aberration correction[17], tissue clearing[18,19], and deep learning[20,21]. To date, these improvements have so far only been demonstrated ex vivo.

Transient absorption microscopy (TAM) is another label-free chemical imaging technique that has found utility in biomedical imaging through the quantification of endogenous heme proteins. Recent reports have used the intrinsic absorption of hemoglobin to quantify hemoglobin glycation[22], oxygenation[23], and concentration[24]. Importantly, imaging of hemoglobin allows direct visualization of blood flow and relieves the need for intravenous fluorescent dye injections into the blood compartment, which are invasive and can suffer from quick clearing times[25]. However, there have been no reports of in vivo TAM imaging of the brain to date.

In this study, we constructed a simultaneous nonlinear absorption, Raman, and fluorescence (SNARF) microscope which combines SRS, TAM, and TPEF for simultaneous imaging of proteins, lipids, hemoglobin, and fluorophores in vivo. Through pulse optimization and image denoising, we performed the deepest reported in vivo SRS imaging of brain tissue at 250–280 μm and the first in vivo TAM imaging of the brain. We employed SNARF microscopy to characterize, quantify, and compare the differences in hemodynamics and cell density between three groups of mice—young (P30–P50), middle-aged (P200–P250),

and old (P630–P635). Using TPEF of a Tie2-GFP transgenic mouse stained with Neurotrace 500/525 as ground truth, we built a Random Forest classifier for label-free identification of two classes of capillary-lining cells—endothelial cells and pericytes—based on nuclear morphology and relation to neighboring capillaries with $90 \pm 2\%$ accuracy. SNARF microscopy builds upon the powerful capabilities of TPEF microscopy by adding orthogonal and complementary chemical contrasts that would otherwise require additional fluorescent labeling. We believe that SNARF microscopy is a comprehensive and versatile in vivo imaging method that will benefit several neuroscience research areas including neurovascular coupling, the blood-brain barrier, cerebral amyloid plaque dynamics, myelin degeneration and regeneration, and neurodegeneration.

## Results

The SNARF microscope is built on top of a commonly used SRS microscope (Fig. 1a) and images multiple contrasts simultaneously (Fig. 1b). In particular, SRS microscopy has been widely used to image the spatial distribution of lipids and proteins in various biological systems ranging from cells to tissue by probing two transitions, 2850 cm$^{-1}$ (CH$_2$ stretching for lipids) and 2930 cm$^{-1}$ (CH$_3$ stretching for lipids and proteins)[9]. Figure 1c, d provides representative in vivo images (~$140 \times 140$ μm) of lipids and proteins, respectively, highlighting key differences in distribution and structure. Figure 1d illustrates that protein distribution is mostly uniform, whereas SRS imaging of lipids in Fig. 1c reveals dark circular features representing cell nuclei and bright thin features representing compact myelin sheaths. Myelin sheaths are multilayer lipid-rich membranes that wrap around axons to provide metabolic support and increase the propagation velocity of action potentials[26]. Both CARS and SRS have been employed previously to visualize and quantify myelination in vivo in the peripheral nervous system, particularly the spinal cord and nerve bundle[27,28]. In vivo CARS imaging of myelin sheath in the brain has been attempted, but the imaging depth was limited to 30 μm[29]. To extract myelin segments from our images, we employed a previously demonstrated tracing program to separate myelin segments directly[30]. Figure 1e provides the resulting image of traced myelin segments. In addition to SRS, the images provided in Fig. 1c, d capture TAM signal of hemoglobin as well. These signals can be separated based on three characteristics: signal intensity, wavenumber dependence, and morphology. TAM hemoglobin signals are stronger than SRS signals due to the larger absorption cross-section. TAM signal is relatively independent of the targeted wavenumber due to the broad absorption profile of hemoglobin. Thus, TAM signal has similar intensity in both Fig. 1c, d. Lastly, tissue structure is nearly static throughout imaging, whereas blood vessels can be identified due to flowing blood cells. These differences allow us to segment blood vessels as shown in Fig. 1f. The raw images were then denoised using a previously described deep learning denoising technique[20,21]. Specifically, the deep learning denoising utilizes a U-Net that was previously trained on low and power SRS image of ex vivo brain tissues to predict high signal to noise ratio (SNR) images from low SNR images in vivo. The denoised images were then spectrally unmixed to obtain a two-color protein and lipid image, similar to previously published stimulated Raman histology approaches[31]. All combined in Fig. 1g, these label-free contrasts provide a snapshot of three cortical structures: cell density, myelination, and microvasculature.

The mouse cortex has six layers containing unique and distinct cellular subtypes that support dynamic brain activity[32]. The outermost layer, layer 1 (L1), has the lowest cellular density of the cortical layers constituting mostly of non-neuronal cells such as astrocytes[33].

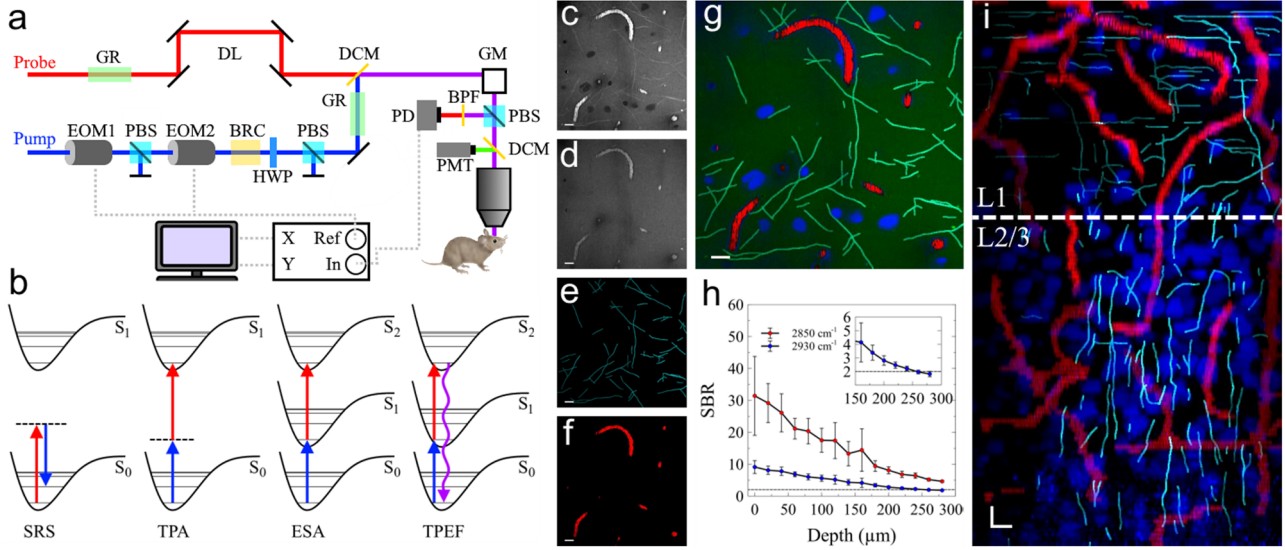

**Fig. 1 Construction of the SNARF microscope and imaging of a live mouse brain. a** Experimental set up. DL delay line, DCM dichroic mirror, GM galvanomirrors, EOM electro-optic modulator, PBS polarizing beam splitter, BRC birefringent crystal, HWP half-wave plate, GS grating-based pulse stretcher, PD: photodiode, PMT photomultiplier tube. **b** Energy level diagrams of the multiphoton techniques employed in SNARF microscopy. SRS stimulated Raman scattering, TPA two-photon absorption, ESA excited state absorption, TPEF two-photon excited fluorescence. **c, d** SRS and TAM imaging at 2850 and 2930 cm$^{-1}$, respectively. **e** Extracted myelin signal. **f** Extracted blood vessels. **g** Composite image of lipids (green), proteins (blue), myelin (cyan), and blood vessels (red) after signal extraction and spectral unmixing. **h** Signal-to-background ratio (SBR) of each wavenumber as a function of depth. Error bars represent standard deviation of all pixels in the images at each depth. **i** Simulated B-slice with a white dashed line at 100 μm below pial surface. Scale bars: 10 μm.

Depending on the specific cortical region, the thickness of L1 ranges from about 100–130 μm. The transition from L1 to layer 2/3 (L2/3) is marked by a sharp increase in cells, primarily neurons, at the border between the two layers. Previous demonstrations of in vivo SRS brain imaging were limited to ~100 μm, suggesting that the L1 to L2/3 transition has never been observed via SRS microscopy[9]. Here, we demonstrate in vivo imaging depths of 280 and ~250 μm for 2850 and 2930 cm$^{-1}$, respectively (Fig. 1h, depth limit is defined at a signal to background ratio of >2), 2.5-fold higher than previous in vivo work. Representative depth images are provided in Figure S1. We achieved this increased penetration depth by employing ~1 ps excitation pulses, which provide higher SRS signals at the cost of decreased spectral resolution. For pulses longer than 1 ps, SRS signal scales inversely with the pulse duration[15]. The use of 1 ps pulses should improve SNR by 7 fold compared to 7 ps pulses used in the previous report, allowing deeper imaging. We measured the spectral resolution of the SNARF microscope to be ~45 cm$^{-1}$ which is sufficient to achieve chemical contrast between lipids and proteins[15]. The deep-leaning-based denoising further improved the imaging depth by ~20%.

With the SNARF microscope, we observed that the number of cells doubled near 100 μm, indicating the transition from L1 to L2/3 (Fig. S2). This trend was observed in all C57BL/6 mice in this study ($n = 10$). Figure 1i shows maximum intensity XZ axis projection from the surface down to 250 μm after separating myelin and hemoglobin signals. From this cross-section, the transition from L1 to L2/3 can be clearly observed by the increase in cell nuclei density around 100 μm below the pial surface indicated by the dashed white line. Additionally, we observed the arborization of myelin sheaths towards the surface of the brain. These structures transition from predominantly ascending, vertical segments at deeper cortical layers to branching, horizontal myelin segments close to the pial surface.

The first cortical structure we investigated was the microvasculature in the top 250 μm of the cortex using TAM and SRS. In vivo TPEF imaging of microvasculature requires an intravenous injection of a dextran-conjugated fluorophore to label the blood plasma[34]. Here, we directly image red blood cells via the strong intrinsic absorption of hemoglobin. Figure S3 illustrates the orthogonal and complementary imaging contrast between TAM and TPEF for in vivo vasculature imaging. Figure 2a provides a 3D rendering of the label-free TAM imaging of a ~140 × 140 × 200 μm³ volume of mouse brain microvasculature. Figure 2b shows the corresponding intensity projection of the volume. Recent studies have demonstrated the occurrence of stalled capillary flow in healthy and disease models, which can lead to decreased oxygen availability and neuronal injury[35,36]. We identify flowing capillaries (shown in red) by the strong RBC contrast and stalled vessels (shown in white) by a noticeable decrease in protein and lipid SRS signal. In our investigation, we observed two classes of stalled capillaries: transient and persistent. The duration of the stall (i.e., transient or persistent) was determined by imaging the same volume of tissue again after 15 min. Persistent stalled capillaries were stalled at both time points, while transient stalled capillaries were only stalled at one time point. Figure 2c, d illustrates the presence of both transiently stalled capillaries (green arrows) or persistently stalled capillaries (purple arrows). The observed prevalence of stalled capillaries for each age group, ~2–5% of capillaries (see Fig. 2e), is lower than reported in the literature for an acute cranial window under anesthesia[36]. This is likely due to our inclusion of only persistently stalled vessel segments rather than all stalled vessel segments. Most stalls were cleared within a short time period[36].

The cause of the stalled capillaries, and decreased cerebral blood flow, has been suggested as a cause for a variety of neurodegenerative diseases[37]. One possible cause of stalls is a narrowing of capillaries which prevents blood cells from passing. We compared the width of 68 stalled vessel segments and 464 flowing vessel segments and found only a slight narrowing for stalled vessel width (Fig. 2f). These results suggest that there is likely another cause of stalled flow. Hernandez et al. demonstrated that

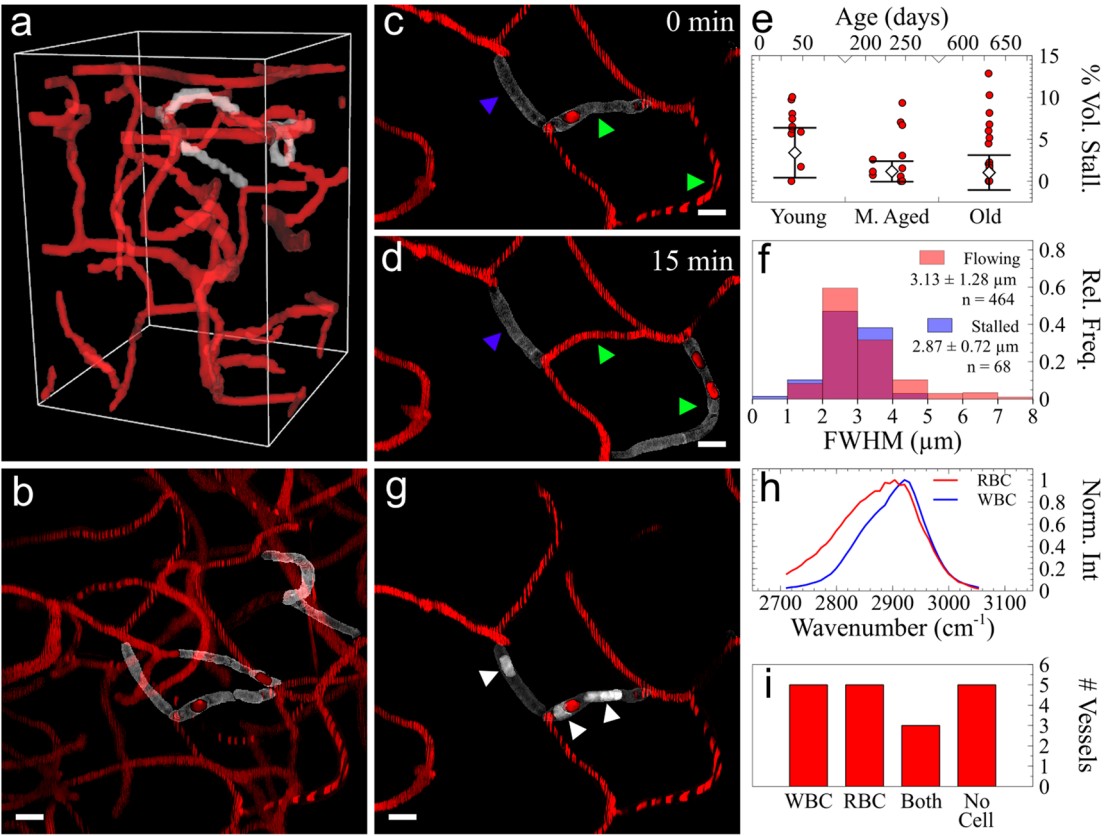

**Fig. 2 Label-free in vivo imaging of mouse brain microvasculature with SNARF. a** 3D rendering of mouse brain microvasculature. **b** Intensity projection of flowing and stalled capillaries from a. **c**, **d** Repeated imaging of vasculature to identify stalled capillaries. **e** Comparison of stalled capillary volume to total vessel capillary between age groups. Error bars represent standard deviation. **f** Average vessel width between flowing and stalled capillaries. **g** SRS imaging of WBCs stalled within vasculature, indicated by white arrowheads. **h** Spectral differences between RBCs and WBCs. **i** Frequency of WBC, RBC, both, or no cell observed in persistent stalled capillaries. Scale bars: 10 μm.

neutrophils plugging capillaries led to decreased cerebral blood flow in an Alzheimer's Disease model[38]. To observe those obstructions, the injection of a fluorescent dye to visualize platelets and white blood cells (WBCs) were required[35,38,39]. Here, we leverage the label-free imaging capabilities of SRS to investigate the occurrence of WBCs within the persistently stalled capillaries. The WBCs, indicated by the white arrows in Fig. 2g, can be separated from RBCs based on intrinsic spectral differences, as shown in Fig. 2h. Stratifying the two types of blood cells allowed us to investigate the contents of stalled capillaries. We observed four categories for the contents of a persistently stalled capillary: (1) only WBC(s), (2) only RBC(s), (3) both WBC(s) and RBC(s), or (4) no cells. Figure 2i provides the frequency of each category over the 18 stalled capillaries identified. Only vessels that were fully contained within the imaging volume were considered. We found that ~45% (8/18 persistently stalled capillaries) contained at least one WBC. Given the low concentration of WBCs within the circulating bloodstream, these data suggest that WBCs have a higher chance of being stuck within, or plugging, a stalled capillary than RBCs. However, a larger number of stalls only have RBCs or no cells, highlighting the need to further explore the extent of stalled flow and underlying causes. Future studies will investigate how transient and persistent stalled flows contribute to local oxygen depletion.

Capillaries are formed by a single layer of endothelial cells. The small capillary diameter helps the efficient exchange of oxygen, nutrients, and waste between the lumen of the capillary and the surrounding tissue. Dilation and contraction of capillary diameter are controlled by pericytes, cells that are embedded within the

walls of capillaries[40]. Both endothelial cells and pericytes are essential for the formation and maintenance of blood brain barrier (BBB) and the regulation of immune cell entry to the central nervous system. TPEF imaging is widely used to study both endothelial cell and pericyte functions in physiological and pathological conditions[41,42]. Either transgenic mice or cell-specific dyes are required to image these cells. Due to the limited color channels (typically 2–3) that can be simultaneously acquired in in vivo TPEF microscopy, the number of cell types and their interactions that can be studied at once is restricted. Here, we perform in vivo SNARF imaging of Tie2-GFP mice to show multiplexing capability and develop label-free identification of capillary-lining cells. Figure 3a provides a 25 μm maximum intensity projection of SNARF imaging of Tie2-GFP in L1 of a Tie2-GFP mouse cortex. The green color represents the lipids and hemoglobin signal, while the blue represents the unmixed protein signal. After segmenting out the capillaries, Fig. 3b shows the cortical cells in blue and the capillaries in red. We topically applied Neurotrace (NT) 500/525 to the exposed cortex before sealing the cranial window to identify the pericytes within the imaged volume. Figure 3c shows the TPEF image of Tie2-GFP (cyan) and NT (green). Tie2-GFP was excited using two-color two-photon excitation of 800 and 1040 nm, and NT was excited using one-color two-photon excitation of 1040 nm. Both dyes were collected using a 525/40 nm bandpass filter and spectrally unmixed after collection.

The first step in label-free cell identification was to stratify each cell identified in SRS imaging into one of two classes: capillary-lining and non-capillary-lining. This was achieved by

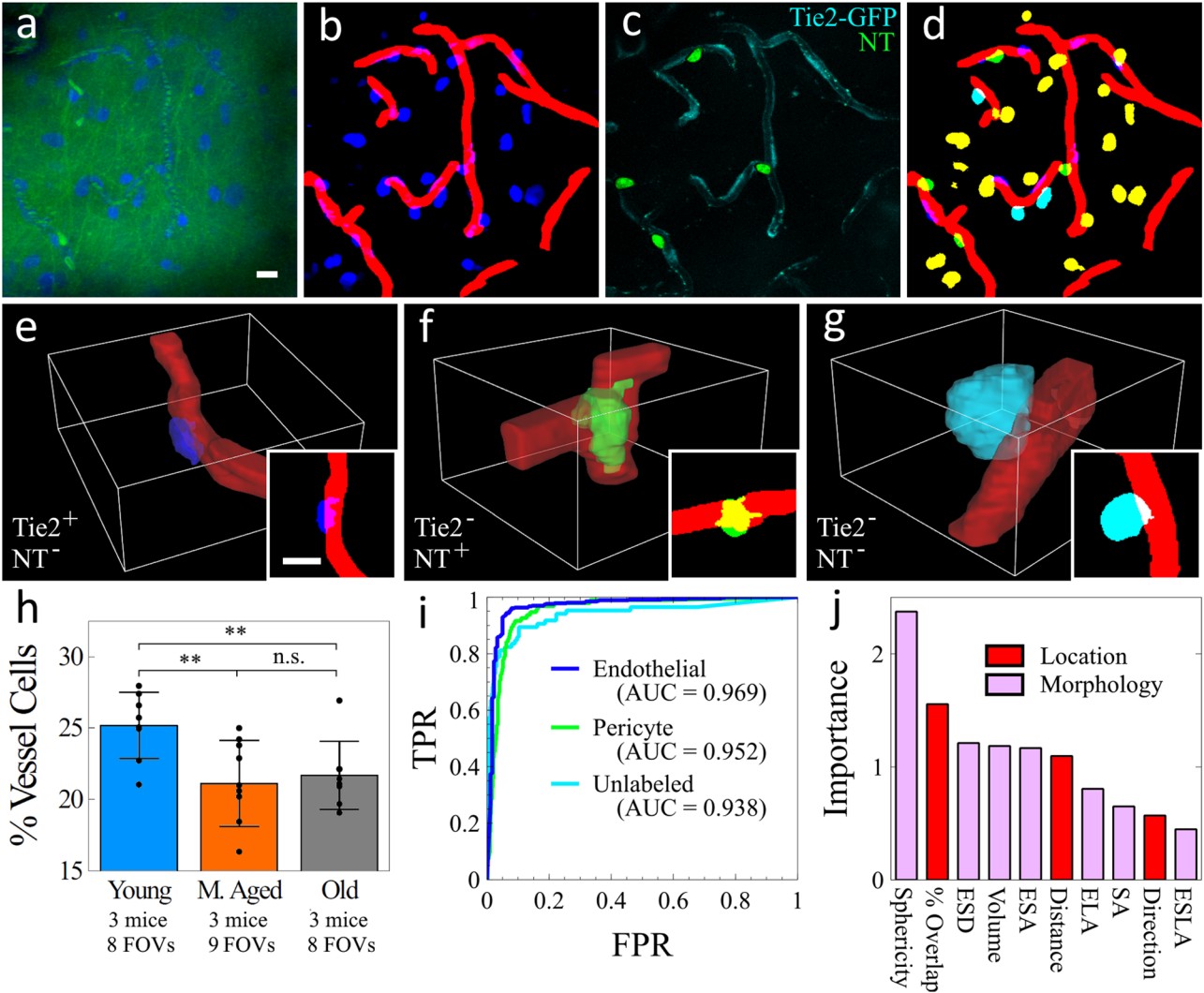

**Fig. 3 In vivo label-free cell identification of capillary-lining cells. a** SRS imaging intensity projection of 25 μm of Tie2-GFP mouse. Green shows SRS of lipids and TAM of hemoglobin. Blue shows the spectrally unmixed proteins. **b** Spectrally unmixed proteins (blue) and capillaries (red). **c** Two TPEF fluorophores: GFP (green) and Neurotrace 500/525 (cyan). **d** Composite image of non-capillary-lining cells (yellow), unlabeled capillary-lining cells (cyan), pericytes (green), and endothelial cells (blue). **e**–**g** 3D rendering of an endothelial cell, a pericyte, and an unlabeled cell, respectively, with corresponding 2D projection. **h** Percentage of vessel cells out of total cell population with age. Error bars represent standard deviation. **i** ROC curves of prediction accuracy for predicting endothelial cells, pericytes, and unlabeled cells based on morphological and locational features. **j** Importance of each feature in the machine-learning classifier. Red features indicate the location of the cell to the capillary, pink features indicate morphology. Scale bars: 10 μm. **\*\*p < 0.01.

determining if any part of a cell was within 1 μm of a capillary. A cell further than 1 μm away was classified as non-capillary-lining cell (color-coded yellow in Fig. 3d), and a cell within 1 μm was classified as a capillary-lining cell and further stratified based on colocalization with the fluorescent labeling. The three classes of capillary-lining cells are endothelial cells (Tie2$^+$, NT$^-$), pericytes (Tie2$^-$, NT$^+$), and unlabeled cells (Tie2$^-$, NT$^-$), color-coded blue, green, and cyan in Fig. 3d, respectively. The "unlabeled cells" class likely contains cell types that can occupy capillary space including astrocytes, oligodendrocytes, and microglia cells. Representative 3D renderings of each cell type are shown in Fig. 3e–g. To understand how aging affects these cell populations, we quantified the number of capillary-lining cells and non-capillary-lining cells between three age groups. Figure 3h shows that the percentage of capillary-lining cells decreases with age which is due to a decrease in capillary-lining cells in age rather than an increase in non-capillary-lining cells as shown in Fig. S4.

After stratifying each capillary-lining cell as an endothelial cell, pericyte, or unlabeled cell, we extracted ten features from each

capillary-lining cell observed. Seven of the features were morphological: sphericity, equivalent spherical diameter (ESD), volume, ellipsoid short axis length (ESA), ellipsoid long axis length (ELA), surface area (SA), and ellipsoid second longest axis length (ESLA). The remaining three features were locational between the cell and neighboring capillary: percent volume overlap (% overlap), distance from cell centroid to capillary, and major axis angle between cell and capillary ($\theta \leq 90°$) (direction). These features were used to train a machine learning model for label-free cell prediction using only SRS images. We achieved a $90 \pm 2\%$ prediction accuracy of 661 endothelial cells, 134 pericytes, and 120 unlabeled cells using a 20-tree Random Forest (RF) algorithm. The RF out-of-bag error plateaued near 20 trees, with negligible improvement from 20 trees to 100 trees (Fig. S5). Figure 3i provides the ROC curves and area under the curve (AUC) for each cell type, and Fig. 3j provides the relative importance of each feature in the RF algorithm. With our trained algorithm, we investigated which capillary-lining cell populations were decreasing with age as described above. We predicted the

pericyte, endothelial, and unlabeled cell populations for three age groups, young (P37–P92, $n = 5$), middle-aged (P209–P246, $n = 3$), and old (P630–P631, $n = 3$), and observed a significant decrease in pericyte and unlabeled cell populations and a slight decrease in endothelial cells with increased age (Fig. S6). These results demonstrate the first in vivo cell identity prediction using SRS morphology. The SNARF method not only improved the multiplexing capability of TPEF, but also enabled label-free imaging of specific cell types through machine learning. These capabilities will facilitate future studies of capillary function in normal aging and dysfunction in neurodegenerative diseases. Importantly, our label-free method can be potentially applied to any mouse which will allow researchers to study disease models without the requirement of dye application or transgenic mice.

## Discussion

The mouse brain is an extraordinarily complex organ with hundreds of different regions and over 100 million cells comprised of many different cell types[1]. Understanding important biological processes of the brain such as neurovascular coupling, the BBB, and neurodegeneration often require the ability to monitor many cell types and features simultaneously in vivo. TPEF has facilitated invaluable contributions to neurophotonics due to its high imaging resolution, versatility, and a wide variety of dyes and transgenic animals. However, despite its tremendous utility, TPEF still suffers from a few key limitations. First, TPEF spectra are typically broad (~50 nm), which can make detection of more than 3 fluorophores in vivo difficult. Spectral unmixing has recently been developed to address this challenge, but it still comes at the expense of system complexity and cost due to the use of multiple lasers and computational unmixing[43]. Second, transgenic animals with multiple fluorescent proteins encoding different cell types require extensive breeding, which brings substantial costs in time and resources. Third, while exogenous dyes can be used to label specific cells or structures, they have also been shown to alter the physiology of the brain. For instance, it has been reported that overexposure of sulforhodamine 101, an astrocyte staining dye, can induce seizure-like brain activity in mice[4]. Additionally, topical application and bolus injection for longitudinal studies become challenging to perform when using a sealed cranial window model. Intravenous injection mitigates the repeated loading problem but only affords a limited time window for imaging and has also shown unwanted side effects. For example, Hernandez et al. reported that intravenous application of a fluorescently labeled antibody used to stain WBCs within a stalled vessel lead to the clearing of stalled vessels 10 min after application[38]. Although TPEF will continue to play a dominant role in high resolution in vivo brain imaging, these limitations highlight the need for alternative imaging methods, such as label-free multiphoton microscopy. The label-free multiphoton imaging techniques—SRS and TAM—described in this study are both orthogonal and complementary to TPEF. All three techniques are coupled into the same laser scanning microscope and share the same excitation laser, which allows for simultaneous detection of multiple contrasts. SNARF offers a richer characterization of brain structure and dynamics. The fluorescence imaging capabilities of SNARF can be further expanded to multiple fluorophores due to the use of two synchronized lasers at different wavelengths. We believe that SNARF multiplexing offers an exciting route to circumvent the limitations of fluorescent labeling.

With the SNARF microscope, we demonstrated the first label-free TAM imaging of cortical hemodynamics. Hemodynamics is a key player in neurovascular coupling and overall homeostasis. We first investigated the recently reported phenomenon of stalled

capillary flow in the cortex. With TAM, we observed three classes of blood vessels: flowing, transiently stalled (<15 min), and persistently stalled (>15 min). With SRS, we further quantified the presence of WBCs within the persistently stalled vessels and found that ~45% (8/18) contained at least one WBC. Others do not have an apparent occlusion or capillary narrowing. Interestingly, we observed no change in the prevalence of stalled capillaries with age. Additional experiments are required to determine if the frequency of stalled WBCs changes in disease conditions. Because stalled flow can lead to local oxygen depletion and has been hypothesized to play a critical role in the development of neurodegenerative diseases, it is important to explore the exact cause of different stalled flow conditions and oxygen delivery change with stalled flow. Although fluorescence tracers are convenient to visualize blood flow, we should exercise caution when using those tracers due to their potential to alter hemodynamics. Another potential benefit of label-free TAM imaging compared to fluorescent tracers is that it is possible to acquire oxygen saturation information based on the difference in absorption between oxy-hemoglobin and deoxy-hemoglobin[23]. This approach could potentially increase the speed of measuring oxygen tension by three orders of magnitude compared to current oxygen probes[44], thus enabling measurement of oxygen delivery change in response to both transient and persistent stalled flow.

Lastly, we used SNARF to quantify and identify the cells that line the capillaries and compose the BBB. With two-color SRS, all the cell nuclei can be visualized, which is the basis for stimulated Raman histology of ex vivo tissue[13]. We quantified the populations of non-capillary-lining cells and capillary-lining cells and found the density of non-capillary-lining cells remained constant with age while capillary-lining cells decreased from young mice (P37–P48) to middle-aged mice (P202–P246). These findings motivated us to build a Random Forest classifier to identify capillary-lining cells based on morphology and relationship to the capillary structure. By using ten features—seven morphological and three relational—we demonstrated 90% accuracy in using label-free SRS imaging to predict three classes of cells: endothelial cells, pericytes, and unlabeled capillary-lining cells. To our knowledge, this is the first report demonstrating the use of SRS to distinguish different cell types in live tissue. We note that it is possible to extend the prediction to other cell types, including astrocytes, oligodendrocytes, and microglia cells, with no additional cost in imaging time or complexity. The prediction accuracy can be improved with larger training datasets, higher SNR, and higher spatial resolution. Both SNR and resolution can be improved through aberration correction as described elsewhere[45]. Deep learning can potentially replace machine learning to offer better performance in cell prediction[46]. In combination with genetic labeling of different subtypes of neurons, SNARF will expand the features that can be monitored simultaneously and provide a powerful platform for in vivo imaging of the structure and function of the brain cortex.

The main limitation of SNARF is the shallow penetration depth of SRS and TAM. Although we have demonstrated the highest reported imaging depth (~300 μm) to date, we are still limited to cortical layer 2/3. In comparison, TPEF can routinely image at a depth of 500–600 μm due to the high sensitivity of fluorescence detection. There are several routes to improve the penetration depth. Aberration correction has been shown to improve the SNR of TPEF in the mouse cortex by five-fold[45]. It can potentially provide a larger benefit for SRS and TAM due to the chromatic and spherical aberrations incurred by employing spectrally separated excitation pulses (i.e., 800 and 1040 nm). Longer wavelength excitation is another strategy that can improve imaging depth for SRS and TPEF[16,47], but it remains to be seen whether TAM signal size will be adversely affected. Because SRS

and TAM use longer pulses than TPEF with less photodamage, it is also possible to use lower rep rate lasers to boost the signal further. Lastly, Raman labels are another possibility to increase SRS signal. While a label is still required in this case, the potential advantage is that Raman labels have much greater capacity for multiplexing than fluorescence labels[48]. We believe that these technical advancements will further improve SNARF microscope to enable multiplex imaging of a wide range of cells, structures, and functions for studying brain function and diseases in vivo.

## Methods

**SNARF microscope system**. The SNARF microscope used for imaging, shown in Fig. 1a, was built upon a previously described SRS microscope system[49]. Specifically, the laser system starts with a femtosecond dual-beam laser (Spectra-Physics, Insight DS+) outputting two beams with an 80 MHz ($f_0$) repetition rate. The two outputs are a fixed beam at 1040 nm, used as the pump, and a tunable beam ranging from 680 to 1300 nm, used as the probe. The pump beam (fixed at 1040 nm) was amplitude modulated at 20 MHz ($f_0/4$) by an electro-optical modulator (EOM1, $\theta_1$) and a polarizing beam splitter (PBS). The amplitude-modulated pulse train was followed by another polarization modulation at 20 MHz with a second EOM (EOM2, $\theta_2$). The second EOM is operated 90° phase shifted from the first EOM (i.e., $\theta_2 - \theta_1 = 90°$), resulting in two orthogonal phase pulse trains (s and p polarization). From here, the beam was passed through a 20 mm of birefringent quartz crystal (BRC) (Union Optic, BIF-Quartz) before traversing a half waveplate (HWP) and PBS to combine both beams into a single polarization. This procedure generates two orthogonally modulated 20 MHz pump pulse trains with a 90° phase shift and a fixed time delay, allowing simultaneous acquisition of two-channel SRS and TAM signals at two different time delays[24,49]. The two pump pulse trains have a measured interpulse delay of 1.1 ps based on the length of BRC used.

Both pump pulses and the probe pulse were sent through 24 cm of highly dispersive glass rod (H-ZF52A) before being spatially overlapped using a dichroic mirror (DCM) and directed through a scanning microscope and 25× objective (Olympus XLPLN25XWMP2 NA = 1.05). Precise temporal overlap was achieved by delaying the probe pulse via a delay line (DL). Fluorescence signal was collected using a long pass DCM and photomultiplier tube (PMT). SRS and TAM signals were collected in epi-mode using a PBS and photodiode (PD). The PD signal was detected by a dual-phase lock-in amplifier (Zurich Instrument HF2LI) with orthogonal outputs after filtering out the two pump beams using two short pass filters (Thorlabs FESH1000). Two images were acquired simultaneously at a frame rate of 1 frame/sec with an image size of 512 × 512 pixels. The spectral resolution of the SNARF microscope was measured to be ~45 cm$^{-1}$. The photophysical processes probed with the SNARF microscope are shown in Fig. 1b.

**In vivo mouse brain imaging**. All experimental animal procedures were approved by the Institute of Animal Care and Use Committee (IACUC) of the University of Washington (protocol # 4395-01). Cranial window surgery was performed on female C57BL/6 mice ranging from P37 to P632 and female Tie2-GFP mice ranging from P31–P92. All mice were purchased from Jackson Laboratory. Briefly, mice were anesthetized using 4% isoflurane (1 L/min O$_2$), and a ~3 mm craniotomy was performed using a high-speed surgical hand drill. Next, Neurotrace 500/525 was topically applied to the exposed brain and a 5 mm coverslip was placed over the craniotomy. Neurotrace 500/525 was applied for 5 min as described previously[50]. Before sealing the window with dental cement (Parkell), the window was filled with modified artificial cerebrospinal fluid to mitigate tissue movement during imaging[34].

**Machine learning metrics and Random Forest algorithm**. The 3D morphology of each cell was calculated using Aivia. The relationship between each cell and neighboring capillary was calculated using Matlab. The distance of each cell to the neighboring capillary was calculated from the 3D center of mass of the cell to the closest edge of a capillary. The direction of each cell was calculated by first segmenting a 5 μm cube centered around the 3D center of mass of the cell. The cell and neighboring capillary contained within the segmented cube were then each modeled as an ellipsoid. The direction was calculated by measuring the angle between the primary axis of the cell ellipsoid and the capillary ellipsoid. A Random Forest (RF) algorithm was employed to distinguish between morphological and locational features of multiple cell classes. We use a 20-tree classification RF algorithm with a 70/30 training/testing separation. The algorithm was trained and tested using Matlab (TreeBagger).

**Statistics and reproducibility**. Statistical significance was determined using a two-tailed unpaired t-test. P values are provided in figure captions.

**Reporting summary**. Further information on research design is available in the Nature Research Reporting Summary linked to this article.

## Data availability
All data are available in the main text or the Supplementary Materials. Data used for the main figures are available in Supplementary Data 1. Any other datasets are available upon request from the corresponding author.

## Code availability
The deep learning denoising algorithm used can be trained using pytorch-fnet U-Net found at https://github.com/AllenCellModeling/pytorch_fnet/tree/release_1 [https://doi.org/10.1038/s41592-018-0111-2][51]. Instructions for training the denoising models are described by Manifold et al. and Hill & Manifold et al.[20,21].

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

## Acknowledgements

The authors would like to thank Dr. Andy Shih for meaningful conversations and guidance with cranial window surgery. The authors would also like to thank the Animal Use Training Services (AUTS) team for the donation of animal tissues that were necessary for the technical development of this project. In particular, the authors would like to thank Kathy Andrich, Erika French, and Francesca Perrotta for their help and training. The study is funded by NIH R35 GM133435 (D.F.), the Beckman Young Investigator Award (D.F.), and the Eli Lilly Young Investigator Award (D.F.).

## Author contributions

A.T.F. and D.F. designed the study; A.T.F., A.H.H., and D.F. developed the methodology; A.T.F. and E.C.C. performed the animal surgery and imaging experiments; A.T.F., B.M., E.C.C., R.H., and S.M. processed and analyzed the imaging data; A.T.F. and D.F. wrote the manuscript with inputs from all authors.

## Competing interests

The authors declare no competing interests.
