## [Peer Review File · Communications Biology]

Reviewers' comments:

Reviewer #1 (Remarks to the Author):

The manuscript reported a combined imaging system that consists of TAM and SRS with TPEF. The imaging depth of SRS reached 250-280 μm . The system was able to show multiple biomarkers, including lipids, proteins, myelin, and blood vessels, due to the combination of imaging modalities. Three age groups were imaged and the results were compared. After quantifying and identifying the non-capillary-lining and capillary-lining cells, the authors used machine learning to predict three classes of cells and reached 90% accuracy, and the prediction is inspiring for other cells. The capability of this system is exciting because it can offer a wide range of data to analyze the metabolism, physiology, and pathology of the animal.

Minor comments.

1. Figure 1A and B were not mentioned until the last section of the manuscript. This misalignment is not friendly, and sometimes confusing, to the readers. Figure 1C-G are imaging the same field of view; however, C-F are much smaller than G. It will be easier for readers to compare the images if they are displayed in the same size. I suggest the authors briefly introduce Figure 1A&B in the beginning of the Results and rearrange the sizes of Figure 1C-G.
2. The blood cell components were analyzed for the stalled capillaries. The authors found that $\sim 45\%$ contained at least one WBC. This is good evidence to support the theory that WBC has a higher chance of being stuck within a stalled capillary than RBC does. However, I was wondering whether the components of the flowing capillaries were analyzed. If so, what is the percentage of the flowing capillaries that have at least one WBC?
3. The Conclusion is more like a Discussion since the authors not only concluded their work but also discussed the pros and cons of a few existing imaging modalities.

Reviewer #2 (Remarks to the Author):

The authors constructed a simultaneous nonlinear absorption, Raman, and fluorescence (SNARF) microscope which combines SRS, TAM, and TPEF for simultaneous imaging of proteins, lipids, hemoglobin, and fluorophores in vivo. The key technical advance is that, through pulse optimization and denoising, they managed to perform the deepest reported in vivo SRS imaging of brain tissue at 250-280 μm and the first in vivo TAM imaging of the brain. To demonstrate this technique, the authors built a Random Forest classifier for label-free identification of endothelial cells and pericytes based on nuclear morphology and relation to neighboring capillaries with 90 % accuracy. This method is general and could find utility in areas including neurovascular coupling, the blood-brain barrier, cerebral amyloid plaque dynamics and myelin degeneration. The paper can be published after the following issues are addressed.

The authors claimed that their deeper imaging ability is built on pulse optimization and denoising. However, the details about this denoising process is lacking. How much does this contribute to deeper imaging?

How much higher SRS signals does the 1 ps pulse result in compared to the previous report? These are important technical issues.

Higher nonlinear photodamage is expected with shorter 1 ps pulse. How does this limit in vivo imaging?

We would like to first thank the reviewers for their thorough reading and constructive comments on our manuscript. We have considered the reviewers commentary carefully and have amended manuscript to reflect the suggested changes. Below, we specifically address the reviewers' comments and how we have accordingly edited the manuscript.

Reviewer #1

The manuscript reported a combined imaging system that consists of TAM and SRS with TPEF. The imaging depth of SRS reached 250-280 um. The system was able to show multiple biomarkers, including lipids, proteins, myelin, and blood vessels, due to the combination of imaging modalities. Three age groups were imaged and the results were compared. After quantifying and identifying the non-capillary-lining and capillary-lining cells, the authors used machine learning to predict three classes of cells and reached 90% accuracy, and the prediction is inspiring for other cells. The capability of this system is exciting because it can offer a wide range of data to analyze the metabolism, physiology, and pathology of the animal.

Minor comments:

1.A. Figure 1A and B were not mentioned until the last section of the manuscript. This misalignment is not friendly, and sometimes confusing, to the readers. Figure 1C-G are imaging the same field of view; however, C-F are much smaller than G. It will be easier for readers to compare the images if they are displayed in the same size. I suggest the authors briefly introduce Figure 1A&B in the beginning of the Results and rearrange the sizes of Figure 1C-G.

We have added references to Figures 1A and 1B to the beginning of the results.

Due to figure size constraints, we have chosen to leave Figure 1G vs 1C-F as it stands. We find that Figure 1G is the most important panel amongst 1C-G with easily separable and identifiable information from each of the nonlinear techniques shown in Figures 1C-F. Figure 1C-F only contains single channel information and has the same size.

1.B. The blood cell components were analyzed for the stalled capillaries. The authors found that ~45% contained at least one WBC. This is good evidence to support the theory that WBC has a higher chance of being stuck within a stalled capillary than RBC does. However, I was wondering whether the components of the flowing capillaries were analyzed. If so, what is the percentage of the flowing capillaries that have at least one WBC?

The mice imaged in this study are normal mice without inflammation. Typical WBC/RBC ratio is 0.1%. Most flowing capillaries have no WBCs. Our imaging throughput is not high enough to provide reliable counts of WBC in flowing capillaries. In addition, due to the imaging slow rate used in this study (<1 Hz), cells are smeared out which degrades imaging quality. This made absolute quantification even more difficult. Thus, we do not provide the percentage in flowing capillaries. In future studies, increasing imaging speed and throughput will be crucial to provide quantification in all capillaries.

1.C. The Conclusion is more like a Discussion since the authors not only concluded their work but also discussed the pros and cons of a few existing imaging modalities.

The reviewer's comment is correct. We have renamed that section of the manuscript to "Conclusion and Discussion" to better reflect our commentary on our technique and experimental demonstrations and future directions.

Reviewer #2

The authors constructed a simultaneous nonlinear absorption, Raman, and fluorescence (SNARF) microscope which combines SRS, TAM, and TPEF for simultaneous imaging of proteins, lipids, hemoglobin, and fluorophores *in vivo*. The key technical advance is that, through pulse optimization and denoising, they managed to perform the deepest reported *in vivo* SRS imaging of brain tissue at 250-280 μm and the first *in vivo* TAM imaging of the brain. To demonstrate this technique, the authors built a Random Forest classifier for label-free identification of endothelial cells and pericytes based on nuclear morphology and relation to neighboring capillaries with 90 % accuracy. This method is general and could find utility in areas including neurovascular coupling, the blood-brain barrier, cerebral amyloid plaque dynamics and myelin degeneration. The paper can be published after the following issues are addressed.

2.A. The authors claimed that their deeper imaging ability is built on pulse optimization and denoising. However, the details about this denoising process is lacking. How much does this contribute to deeper imaging?

We have added additional description of the deep learning denoising process to the introduction sentence referenced by the reviewer. Hill & Manifold et al (Ref. 21) demonstrate a 24% improvement (165 μm to 210 μm) in imaging depth based solely on deep learning denoising in fixed tissue. We utilized the same denoising algorithm in addition to our different pulse optimization to image *in vivo* down to depths of 250-280 μm . Without the denoising, we were able to observe reliable SNR and features down to ~200 μm .

2.B. How much higher SRS signals does the 1 ps pulse result in compared to the previous report? These are important technical issues.

The relationship between pulse duration/bandwidth and SRS signal has been shown theoretically and experimentally in the author's previous study (see <https://doi.org/10.1371/journal.pone.0178750>) as well as several other sources cited within. For pulses longer than 1ps, the signal is inversely proportional to pulse duration. The use of ~1 ps pulses compared to 7 ps pulses (as used in the earlier SRS studies with picoEmerald laser) should provide 7-fold improvement in signal. Longer pulses are typically used for high spectral resolution. Here, we make use of the broad CH stretching of proteins and lipids to lower our spectral resolution in favor of shorter pulse durations, and thus higher signal sizes. We add some more details to the manuscript to explain this issue.

2.C. Higher nonlinear photodamage is expected with shorter 1 ps pulse. How does this limit *in vivo* imaging?

The reviewer correctly notes the potential nonlinear photodamage increase for shorter pulses. As suggested by Podgorski & Ranganathan (doi:10.1152/jn.00275.2016), duty cycle and effective illumination time matters significantly with respect to onset of irreversible tissue damage in multiphoton imaging schemes. For example, they suggest lasting histological damage is observed in awake mice at 250-300 mW continuous illumination for 20 minutes at 800 nm with ~120 fs pulse durations using a high NA objective. Taking their suggested threshold and designing our experiment conservatively, we utilize 150 mW total beam powers (50 mW in each beam) at 1 ps for max illumination time of ~4 min (e.g. for a given depth stack). The nonlinear photodamage study by Hopt and Neher showed that photodamage scales with the integral of light intensity raised to a power ≈ 2.5 while signal scales with I^2 . Therefore, compared to femtosecond lasers, picosecond lasers should have much lower photodamage if we use

equivalent average power. In our experiment, we have not noted any observable tissue damage during our experiments.